# Asthma with Fixed Airflow Obstruction: From Fixed to Personalized Approach

**DOI:** 10.3390/jpm12030333

**Published:** 2022-02-23

**Authors:** Agamemnon Bakakos, Stamatina Vogli, Katerina Dimakou, Georgios Hillas

**Affiliations:** 11st University Department of Respiratory Medicine, National and Kapodistrian University of Athens, 11527 Athens, Greece; 25th Pulmonary Department, ‘Sotiria’ Chest Diseases Hospital, 11527 Athens, Greece; stamvog95@gmail.com (S.V.); kdimakou@yahoo.com (K.D.); ghillas70@yahoo.gr (G.H.)

**Keywords:** asthma, fixed obstruction, personalized medicine, ACO, FAO

## Abstract

Asthma is generally characterized by variable symptoms such as dyspnea and wheezing and variable airflow obstruction. This review focuses on a subset of patients suffering from asthma with persistent airflow limitation that is not fully reversible (asthma with fixed airflow obstruction, FAO). The pathophysiology, the risk factors and the clinical outcomes associated with FAO are presented, as well as the distinct clinical entity of severe asthma and its inflammatory subtypes (T2 and non-T2). The current strategies for the treatment of these endotypes and treatment of the distinct Asthma/COPD overlap (ACO) phenotype are described. Management and medical interventions in FAO and/or ACO patients demand a holistic approach, which is not yet clearly established in guidelines worldwide. Finally, a treatment algorithm that includes FAO/ACO management based on pharmacological and non-pharmacological treatment, guideline-based management for specific co-morbidities, and modification of the risk factors is proposed.

## 1. Fixed Airflow Obstruction: Is It Due to Asthma, COPD or Asthma-COPD Overlap?

Although airflow obstruction that is variable and reversible is a defining feature of asthma, subgroups of patients with long-term persistent asthma may develop fixed airflow obstruction (FAO). Traditionally, FAO characterizes chronic obstructive pulmonary disease (COPD) with the definitions of asthma and COPD not being mutually exclusive [1]. The possible progression of persistent asthma into FAO is consistent with observations showing that, at the population level, co-existing diagnoses of asthma and COPD are frequently reported by the same patient [2] and that active persistent asthma substantially increases the risk of acquiring a subsequent diagnosis of COPD [3]. “Asthma-COPD overlap” (ACO) is a descriptor for patients that are assigned both diagnoses [1] with prevalence rates ranging between 9% and 55% of those with either diagnosis. The wide range reflects the variation by gender and age as well as different criteria that have been used by different investigators [3,4]. ACO diagnosis should involve objective and specialized examinations, in order to distinguish between asthma and COPD. Lung function tests such as diffusing capacity of the lung for carbon monoxide (DLco), imaging such as high-resolution computed tomography (HRCT) and inflammatory biomarkers such as specific-immunoglobulin E (IgE) and fractional exhaled nitric oxide (FeNO) have been widely used for this purpose. In a study that assessed the prevalence of both FAO and COPD components in elderly asthma patients, DLco% predicted <80% and the appearance of low attenuation area as a percentage (LAA%) in HRCT were used as COPD markers. The prevalence of asthma patients with FAO increased with age and as a whole, 45% of the patients older than 50 years of age showed FAO. Nearly half of asthmatics had findings compatible with coexisting COPD, with much higher prevalence in ex- and current smokers than in never smokers, despite the almost same prevalence of FAO. These findings show that smoking plays a major role in ACO development, especially in male patients, and can aid clinicians distinguish between asthma with FAO and ACO [5].

The differential diagnosis between FAO due to asthma or COPD can be quite difficult but is crucial in clinical practice, because the prognosis and the response to treatment of the two diseases are different. Specific features of history and clinical assessment are of great importance in identifying and distinguishing asthma from COPD and spirometry plays a major role in confirming the presence of fixed or variable expiratory airflow obstruction [1]. Interestingly, measurement of lung volumes, responsiveness to bronchodilators or corticosteroids, and even diffusing capacity overlap considerably, making these tests of little use for distinguishing asthma with FAO from COPD. Airway responsiveness to methacholine has also been found not to be significantly different between the two groups, confirming that, once FAO develops, measurement of airway hyperresponsiveness is not useful any more [6,7].

Currently, there is a variety of definitions of FAO used in clinical practice and medical research including use of pre- and/or post-bronchodilator spirometry values, Forced Expiratory Volume in the 1^st^ second/Forced Vital Capacity ratio (FEV_1_/FVC), absolute or relative cutoffs as well delineation based on the absence of a bronchodilator response alone. Spirometry predominantly assesses the flow of larger more central airways and is not a sensitive measure of the small peripheral airway dysfunction and abnormal mechanical properties of lung tissue that contribute to the pathogenesis of severe asthma [8,9]. In order to assess the small airways and parenchyma in asthma, a number of different techniques have been used [8,9,10]. Oscillometry, increasingly utilized both in clinical practice and medical research, does not require effort to force expiration that may affect small airway closure and can differentiate if an increase in the total resistance is located at central or at peripheral airways. The prevalence of increased small airway resistance using oscillometry has been found to be significantly higher in asthmatic patients with FAO and COPD than in asthmatics without FAO [9]. In addition, the presence of abnormal post-bronchodilator reactance in oscillometry has been related to low asthma control in asthma patients with FAO and contributed in identifying 25% more patients with poor asthma control compared to spirometry parameters alone [10].

The prevalence of asthma history among patients with FAO has been reported up to 30% [11]. A number of studies have compared airway inflammation patterns in predefined patients with either asthma or COPD who have developed FAO [12,13,14]. In asthma, airflow obstruction is mostly associated with a characteristic airway inflammation consisting of an increased number of T lymphocytes (predominantly CD4+) and eosinophils and an increased thickness of the reticular layer of the epithelial basement membrane. Contrarily, FAO related to COPD is associated with an inflammatory profile consisting mainly of T lymphocytes (predominantly CD8+), macrophages, and neutrophils. Asthma has also been associated with neutrophilic inflammation. Neutrophilic inflammation in asthma is less common and is mostly driven by lifestyle or environmental factors, such as obesity, smoking, late age of onset and occupational exposures. Although sputum eosinophilia is one of the most important determinants of the development of FAO in severe asthma, marked airway neutrophilia has been found to correlate with FAO in asthma as well [15,16]. However, the exact impact of neutrophilic lung inflammation in FAO development is unclear, since other studies have not demonstrated differences in sputum neutrophils between asthmatics with and without FAO [17]. Even after the progression of FAO, patients with a history of asthma may continue to exhibit a distinct airway inflammation as compared with those with a history of smoking-induced COPD [18].

Consistent with findings of subsequent studies, Fabbri et al. [18] first indicated that asthmatic patients with FAO had lower residual volume, higher diffusing capacity and oxygen tension in arterial blood, and an increased response to inhaled albuterol and steroids compared to patients with COPD and the same degree of FAO. Moreover, patients with a history of asthma had more eosinophils in peripheral blood, sputum, bronchoalveolar lavage fluid and airway mucosa and had fewer neutrophils in sputum and bronchoalveolar lavage fluid. In terms of pathology, asthmatic patients had more bronchoalveolar lymphocytes and more CD4+ cells, a higher CD4+/CD8+ ratio and a thicker reticular layer of the basement membrane compared with patients with same degree of FAO diagnosed as a consequence of smoking-related COPD. The relationship between airway inflammation and airflow obstruction is not fully understood both in asthma and in COPD. The increased number of a certain cell type is unlikely to be the major determinant of airflow obstruction, but is rather a marker of two different inflammatory cascades for two different diseases, asthma and COPD, which result in the same functional abnormality, i.e., FAO.

## 2. Pathophysiology of FAO

The structural changes of the airways, broadly referred to as airway remodeling, is universally perceived as a leading cause of FAO in asthma, but the exact underlying mechanisms are yet to be elucidated. Cellular and extracellular matrix changes in the large and small airways, epithelial cell apoptosis, airway smooth muscle cell proliferation, and fibroblast activation are some of the key characteristics of airway remodeling [19]. Although eosinophils and CD4+ lymphocytes seem to play important roles in directing and maintaining the inflammatory cascade, the association between remodeling and inflammation is still unclear. Allergen-induced myofibroblasts in the airways of asthmatic patients have been found to persist even after the dissolution of inflammatory infiltrates. Mediators like bradykinin may provoke acute inflammatory effects in the airways, including induction of fibroblasts to differentiate into α-smooth muscle actin (α-SMA)+ myofibroblasts and release proangiogenic factors, thus contributing to airflow obstruction in chronic asthma. Patients with severe asthma demonstrated high expression of bradykinin receptors B1 and B2 (B1R, B2R) in their bronchial submucosa, in the α-SMA+ mesenchymal cells of the bronchial mucosa and in their epithelium as well. Interestingly, α-SMA and B1R were markers associated with impaired pulmonary function and development of FAO [20].

Apart from airway remodeling, evidence from studies [21,22,23] in asthmatic subjects with obstructed spirometry suggests that structural and functional changes in the lung tissue also likely contribute to FAO. Lung tissue abnormalities may reduce lung elastic recoil, thereby reducing the outward recoil to the airways, which maintains airway caliber and alveolar, driving pressure during exhalation. Indeed, reduced lung elastic recoil pressure can reduce the airway caliber at any given lung volume, increasing airway resistance and airway closure, both of which would contribute to airflow obstruction [24]. Gelb et al. [22] reported reduced elastic recoil pressure at 70% and 80% of total lung capacity (TLC) and reasoned that reduced lung recoil pressure at a single lung volume point would partly account for reduced airflow. Whether lung tissue abnormalities have distinct functional roles, irrespective of airway disease, in asthma pathology and FAO development is yet to be elucidated. There is evidence that parenchymal destruction occurs in asthmatics with FAO regardless of smoking status and asthma severity. In a study of asthma patients, the usage of computed tomography revealed decreased lung density (as reflected by decreased exponent D) and an increase in LAA%, which is used as an emphysema index. Moreover, the exponent D was lower and LAA% was higher in asthmatics with FAO than in those without FAO regardless of smoking status, pack-years, age, sex, BMI, asthma severity, atopy, and blood eosinophil count [25]. These data are suggestive of the key role that parenchymal destruction may play in asthma with FAO, affecting the longitudinal decrease of lung function in both smokers and non-smokers.

## 3. The Distinct Clinical Entity of Severe Asthma

The structural and functional changes of the airways and, to a lesser extent, the lung tissue leading to FAO, are pathological features present mostly in severe asthma and contribute to the clinical manifestations of the disease. Severe asthma is a subset of difficult-to-treat asthma and is defined as asthma that is uncontrolled, despite adherence with maximal optimized high dose inhaled corticosteroid–long acting beta-2 agonist (ICS-LABA) treatment and management of contributory factors, or that worsens when a high dose treatment is decreased [1] and it is estimated that around 3.7% of asthma patients have severe asthma [26]. Apparently, severe asthma is a different disease than mild and moderate asthma. Patients with severe asthma have symptoms which are difficult to control, require high dosages of medication, and continue to experience persistent symptoms, asthma exacerbations or FAO, even with aggressive therapy. The prevalence of FAO among patients with severe asthma is estimated to be 55% to 60% [27].

With regard to pathogenetic mechanisms, dysregulation of the T-helper cell type 1 (Th1) and Th2 cytokine production in severe asthma has been shown to differ from mild asthma, but the exact underlying mechanisms in a number of severe asthma phenotypes are not well defined yet [28]. Eosinophils are still thought to be important effector cells in severe asthma, but increased airway neutrophils have also been found in a portion of patients with severe, persistent asthma [29].

The field of personalized medicine in asthma management has benefitted greatly from the recognition that “severe asthma” refers to an umbrella term encompassing a range of clinical phenotypes. Different phenotypes of severe asthma have been proposed based on various patient characteristics such as lung function, age of onset of asthma, atopic status, sputum eosinophil and FeNO [30,31,32]. Several studies have applied these parameters with clustering algorithms to asthmatic patient cohorts, resulting in identifying a number of distinct clinical clusters [33,34,35]. Eventually, a key goal of this phenotyping process is a linkage to molecular mechanisms, defining an endotype, which would predict a response to therapy. Based on the presence of T2-driven inflammatory responses, two major asthma endotypes, T2 and non-T2, have been described.

## 4. Inflammatory Subtypes of Severe Asthma: T2 and non-T2

Asthma is traditionally thought to result from inflammation driven by T2 high responses and mediated by cytokines including IL-4, IL-5, and IL-13, which are often produced after the recognition of allergens. Specifically, IL-13 is produced by activated T cells, basophils, eosinophils, and mast cells and is thought to be a central molecular mediator of inflammation in asthma. T2 inflammation may also be activated by viruses, bacteria and irritants that stimulate the innate immune system via the production of IL-33, IL-25 and thymic stromal lymphopoietin (TSLP) by epithelial cells [36]. It is often characterized by blood and sputum eosinophilia or increased biomarkers associated with eosinophilic activation such as FeNO, serum periostin, serum eosinophil cationic protein (S-ECP), and urinary eosinophil-derived neurotoxin (U- EDN). Increased expression of tumor necrosis factor a (TNF-α in macrophages in T2-high severe asthma has also been described, despite it not being considered a T2 cytokine [37]. T2-high asthma is usually accompanied by atopy, although Woodruff et al. [38] found that both T2-high and T2-low subgroups of asthmatic patients had increased allergen skin prick test reactivity as compared with healthy control subjects, with the T2-low subgroup having fewer positive tests than the T2-high subgroup. Diagnostically, the possibility of severe-refractory T2 high asthma should be considered if any of the following are found while the patient is taking high dose of corticosteroids: blood eosinophils ≥ 150/μL; FeNO ≥ 20 ppb; sputum eosinophils ≥ 2%; asthma that is clinically allergen-driven [1].

Several studies have investigated the relationship between markers of T2 inflammation and severity of asthma or development of FAO. Blood eosinophils and FeNO have been found elevated in asthmatics with FAO, and together with serum periostin, are markers linked to decreased lung function [39,40]. Positive correlation between levels of the eosinophil activation marker U-EDN and FAO has been found with the association being greater in subjects who had a simultaneous elevation of both markers, S-ECP and U-EDN [37,41]. On the other hand, atopic status is less common in asthmatics with FAO, although the correlation between eosinophilic markers and FAO has been strongly established [17]. This demonstrates that the actual endotype leading to eosinophilic inflammation affects the likelihood of developing FAO. Innate lymphoid cells type 2 (ILC-2) producing IL-5 and IL-13 could be one such factor, driving a T2-high inflammation dependent on IL-5, with eosinophil predominance and possibly a site of inflammation either systemic or located to the smaller airways, less sensitive to ICS. This is contrary to high IL-4/IL-13 inflammation located to the bigger airways or airway mucosa, sensitive to ICS [37].

In a longitudinal study among adults with moderate-to-severe asthma, the incidence of FAO after 8 years was 36.4% and two different or sequential FAO endotypes, long-term persistent and long-term incident, were reported. These distinct endotypes exhibited different clinical and biologic features with the long-term incident FAO group having a higher sputum eosinophil content, baseline FEV_1_ variability and longitudinal FEV_1_ decrement, but lacked evidence of baseline airway remodeling in terms of increased airway smooth muscle area. On the other hand, the long-term persistent FAO group exhibited a higher sputum neutrophil content, evidence of airway remodeling and lower FEV_1_, whereas the long-term change in FEV_1_ was small, and similar to the group without FAO [42]. Interestingly, pathologic changes in severe T2-high asthma with airway remodeling may differ, with increased reticular basement membrane thickness, a finding not common in neutrophilic asthma [38]. Evidence that severe asthma and FAO development may be associated with sputum neutrophilia raises questions about the role that neutrophils and non-T2 mediated inflammation might play in FAO [16].

Non-T2-high asthma is characterized by the absence of markers of T2 inflammation, neutrophilic or paucigranulocytic (absence of sputum eosinophilia or neutrophilia) cellular phenotype and Th1 and/or Th17 being the key effector cells [29]. Airway neutrophilia accounts for 5–22% of asthmatic patients, often related to more severe disease with worse pulmonary function, poor asthma control and FAO [15]. The neutrophilic phenotype may be associated with pollution, smoking, workplace agents, acute and chronic infections and is common in smokers with asthma, obese asthmatics and occupational asthma [43]. It is characterized by a neutrophilic proportion of ≥64% in induced sputum and does not correlate with blood neutrophiles, in contrast to blood and sputum eosinophilia [44]. Important markers of neutrophilic induction are IL-6/IL-8, as well as myeloperoxidase and neutrophil elastase. Another phenotype under the term “non-T2 asthma” is asthmatic patients with both neutrophil levels < 64% and eosinophil levels < 2%, who are classified as paucigranulocytic asthmatics. A substantial proportion of them (21.7%) have been reported to have severe refractory asthma [45]. This phenotype is in concordance with results from animal studies proposing that processes stimulating airway remodeling with smooth muscle thickening occur independently from inflammation and are a consequence of a loss of negative homeostatic processes [46].

## 5. Risk Factors and Clinical Outcomes Associated with FAO in Asthma

In order to recognize and evaluate the risk factors for FAO development, several studies have been performed but with significant heterogeneity concerning inclusion and exclusion criteria as well as their conclusions [27,34,47]. Among the most commonly proposed risk factors are blood and sputum eosinophilia, adult-onset asthma, older age, male sex, and a history of smoking [17]. Airway hyperresponsiveness has also been reported but results from different studies are inconsistent. Likewise, paradoxical associations have been noted with allergic rhinitis, both the presence and absence of which have been highlighted as potential risk factors for FAO [17,47]. Differences in proposed risk factors and associations may be related to differences in populations, and this demands further analyses in order to conclude to robust results. A study focused on older asthmatic patients with FAO found as independent risk factors male sex, increased duration of asthma, older age, and African American race [48]. In an attempt to investigate the association of FAO progression with ageing, Guera et al. [12] found that aging was strongly associated with airflow limitation both among subjects with and without asthma. The profile of risk factors for FAO was dependent upon the presence and the age at onset of asthma, with blood eosinophilia being the strongest risk factor for subjects with asthma onset before 25 years of age. As regards smoking, lifetime exposure to tobacco was associated with an increased risk for FAO that was almost twice as high among subjects with asthma onset >25 years of age than among subjects with asthma onset <25 years of age. Obesity and BMI are characteristics usually assessed in published studies on asthma with FAO but no independent association has been reported yet. Interestingly, a higher BMI has been associated with less airflow obstruction in asthmatics, a finding potentially mediated through BMI-related mechanisms, which increase lung stiffness. This study confirms the hypothesis that obesity interacts with asthma in a complex way and affects the lung elastic properties [49].

Despite the abundance of the proposed risk factors for FAO development, the latest update of GINA report has incorporated the following FAO predictors: history characteristics (preterm birth, low birth weight and greater infant weight gain); chronic mucus hypersecretion; lack of ICS treatment in patients who had a severe exacerbation; tobacco smoke; noxious chemicals; occupational exposures; low initial FEV_1_; sputum or blood eosinophilia [1]. The possible associations of early growth characteristics with childhood and adult asthma might be explained by the hypothesis that developmental adaptations of the lungs and airways may lead to relatively small airways and hence a reduction in expiratory flow, with preterm birth and low birth weight resulting into a persistent reduction of airway patency [50]. Although the precise risk factors for FAO in children remain mostly unknown, there is evidence they are quite similar to those observed in adults. A prospective cohort study with asthmatic children and adolescents revealed that 9.5% of them developed FAO during the 4-year follow-up. Subjects in FAO group were older than 10 years of age, had a higher BMI, more severe asthma and reported higher budesonide consumption and more frequent exacerbations and hospitalizations than those in the non-FAO group. Among the investigated characteristics, only frequent exacerbations and asthma severity categorized as steps 4 to 5 were proven independent risk factors for FAO [51].

Except for the differences in clinical and inflammatory traits, patients with FAO have been found to exhibit different functional characteristics and disease outcomes compared with asthmatic patients without FAO. Significantly longer disease duration, lower FEV_1_, FVC and FEV_1_/FVC ratio and lower PD_20_ values are present in asthmatic patients with FAO. They may, also, have an increased comorbidity, a greater risk of intubation and a higher mortality [52]. Their subjective asthma scores and disease-related quality of life have been reported to differ with worse disease course when FAO is present, especially in younger patients [27]. The natural course of the disease and 5-year longitudinal data of clinical outcomes reveal that asthma patients with FAO exhibiting a progressive postbronchodilator FEV_1_ decline over time with a rate similar to this of patients with same degree of FAO at the beginning of follow-up due to COPD. The mean number of exacerbations per patient-year was no different between patients with FAO due to asthma or COPD and it was significantly higher compared with that seen in the asthmatic patients without FAO. In addition, sputum eosinophils at baseline were found to corelate positively with the frequency of exacerbation in patients with FAO due to asthma [53].

The aforementioned points underline that asthma with FAO is a complex disease with great heterogeneity and unidentified pathogenetic mechanisms. Further isolation of risk factors and identification of phenotypes and endotypes will contribute to a more personalized medical approach for optimal evaluation and treatment options.

## 6. The Non-T2 High FAO Endotype: The “Fixed” Treatment FAO Asthma

Although hard to measure accurately, FAO is more common as the severity of asthma increases [54]. Therefore, FAO+ asthmatics usually need a combination of ICS/LABA to be relieved from symptoms, rather than monotherapy.

Initial evidence supporting this theory of more “aggressive” approach was derived from a post hoc analysis of two randomized control trials (RCTs) in 2014 by Tashkin et al. [55], where patients were randomized to a combination of budesonide/formoterol (BUD/FM), budesonide or formoterol alone and placebo. FAO+ patients did not exhibit any benefit by the use of FM only, since their lung function was similar to the placebo group, but interestingly the combination of BUD/FM in these patients was far more effective than BUD alone in improving lung function and outcomes, such as the use of rescue medication and awaking during sleep. This shows a possible synergy of the combined treatment with ICS/LABA in an exponential, rather than additive way. Similar data have also been reported in vitro for the synergy of double therapy with ICS/LABA [56,57].

The same research team managed to further underline this point two years later by randomizing moderate to severe asthma patients to a combination of BUD/FM, BUD alone, FM alone and placebo. The key difference of this work was that they determined the patients’ FAO status not only at enrollment but also during the trial. This revealed an inconsistent FAO group, meaning that at some point patients did express FAO+ features, but not constantly. The interesting detail was that these asthmatics responded to treatment very similarly to the constantly FAO+ group. The combined therapy was found to be more potent, whereas other treatment options were related to more worsening-asthma events and worse lung function in the inconsistent and FAO+ groups. Moreover, the double therapy group was less likely to progress from FAO–to FAO+ during the trial, a risk that was more elevated in the other arms [58].

The reduced responsiveness of FAO+ asthmatics to LABA and ICS treatment has raised interest as to whether this could be attributed to a dysfunction of their airway smooth muscle cells. Researchers used bronchial biopsies to acquire airway muscle cells from severe asthmatics with FAO, mild asthmatics without FAO and healthy volunteers and then stimulated them with either salbutamol or fluticasone. Although the trial included a very small number of patients, the in vitro response of the cells to LABA and ICS was not different in these three categories, which consequently means that most likely FAO is not a result of dysregulated adrenergic or corticosteroid signaling in smooth muscle cells [59].

A combination of inhaled corticosteroids with long-acting muscarinic antagonists (ICS/LAMA) that was tested in a phase III RCT is the fluticasone furoate-umeclidinium combo (FF/UMEC). The combination with FF dosage being 100 mcg and UMEC dosage being 62.5 mcg was compared to treatment with FF alone, FF-vilanterol combo as well as FF with other UMEC doses in patients with FAO and a diagnosis of either asthma or COPD. Once again, the double therapy, this time including LAMA instead of LABA, was superior compared to ICS monotherapy in increasing FEV_1_ (*p* = 0.019) for the 62.5 mcg UMEC dose combo. Statistically important results were also observed with all FF/UMEC combinations vs. monotherapy in terms of morning peak expiratory flow and, moreover, patients under combination therapy showed a significant reduction of usage of rescue therapy [60].

Bronchial thermoplasty (BT) is another treatment option for severe refractory to treatment asthma. The logic behind this procedure is that it counters one of the hallmarks of bronchial asthma, airway remodeling. This exact feature eventually leads to FAO, since hypertrophied airway smooth muscle cells respond less efficiently to bronchodilation therapy [61]. BT uses radiofrequency thermal energy, which in turn leads the smooth muscle cells to atrophy. In a prospective study in Australia, 49 patients who underwent BT were spirometrically evaluated before and after the intervention, according to whether or not they expressed FAO. Both groups responded equally well to BT and after a 6-month follow up researchers observed important improvements in patients’ quality of life (QoL), exacerbation rate and weaning of oral corticosteroids (OCS) [62]. Although not a first line choice, BT should be considered a viable choice for all severe asthmatics, FAO+ equally included.

An interesting concept for FAO+ asthmatics is its co-existence with high BMI. Obesity is a common comorbidity in asthmatics. Fixed obstruction is associated with higher lung compliance and thus reduced lung stiffness (elastance), which also acts as a predictor of the severity of the obstruction [23]. On the other hand, obesity is associated with lower compliance and higher elastic recoil of the respiratory system [63]. At first glance, it looks attractive to contemplate whether obesity could actually act “protectively” in FAO+ asthmatics. Indeed, researchers have found in a small study including 18 asthmatics that a higher BMI increased the elastic properties of the lungs and reduced airflow limitation, although it had no effect on FVC and TLC [49]. However, this is by no means an indication that FAO+ patients should remain obese, since it is also well established that obesity alters the response of smooth muscle cells in the airways by inducing inflammation and can lead to bronchoconstriction even in healthy subjects without asthma [64]. Furthermore, several clinical trials have demonstrated that weight loss in obese patients can ameliorate asthma symptoms and even lead to a reduction in doses of rescue medication [65,66]. Do FAO+ patients comprise a different category? This complex relationship between FAO and high BMI has yet to be elucidated.

The neutrophilic endotype is the most difficult to treat endotype up to date, since we have no targeted therapy or treatable traits, only a generalized approach. Nevertheless, a key concept that can greatly improve patients’ quality of life should be underlined. It is established that smoking cessation is essential in reducing symptoms of neutrophilic inflammation and improve lung function, regardless these patients have fixed obstruction or not [67]. Patients should be encouraged to quit smoking and get referred to a specialized smoking cessation program in order to improve their chances of success [68].

Conclusively, with the knowledge that persistent FAO+ is heavier linked with neutrophilic inflammation, in contrast with inconsistent FAO, which is associated with eosinophilic inflammation [42], we can attempt to personalize our treatment options based on clinical features and biomarkers, especially in the setting of T2 high endotype. Although the neutrophilic T2 low endotype has few personalized options, our arsenal is abundant with weapons for the T2 high endotype.

## 7. The T2 High FAO Endotype: An Ode to Personalized Medicine

It has been established that among asthmatics with FAO, eosinophilic inflammation may attain a crucial role in the development of non-reversible obstruction and deterioration of lung function. With this fact in mind, it is fascinating to speculate as to whether eosinophils and T2 inflammatory cytokines are treatable traits of the disease and might respond to targeted therapy [37].

Mepolizumab, an anti-IL-5monoclonal antibody (mAb), was initially evaluated in two RCTs, the DREAM [69] and MENSA [70] trials for use in severe eosinophilic asthma. Later on, experts utilized data from these trials and divided asthmatics based on COPD-like features, namely an age > 40 years old, FEV_1_/FVC < 0.70, a post bronchodilator FEV_1_ < 80% predicted and smoking history. Mepolizumab effectively reduced the annual exacerbation rate (AER) in asthmatics with fixed obstruction as well as asthmatics who were former smokers, showing that eradication of eosinophilic inflammation could aid not only severe asthmatics with FAO but even patients in the COPD spectrum with eosinophilic infiltration of their lungs [71].

Similar results were recently published by Pavord et al. [72] in a meta-analysis of COPD patients under triple therapy with signs of eosinophilic inflammation (at least 150 eosinophils/μL at screening or at least 300 eosinophils/μL during the previous year). Mepolizumab was efficient in reducing the AER of these patients and by direct extrapolation of these results on asthmatics with FAO, although the underlying disease is not the same, we can assume at least the same effectiveness in terms of controlling the burden of eosinophilic infiltration.

A pooled post hoc analysis of SIROCCO [73] and CALIMA [74] phase III trials of the anti-interleukin-5 receptor (IL-5R) mAb benralizumab, stratified severe asthmatics based on whether they had features of fixed airflow obstruction or not. The threshold used was an FEV_1_/FVC ratio < 0.70 post bronchodilation. Out of a total of 1493 patients, 63% were FAO+. Although the reduction of the annual exacerbation rate was similar to the FAO+ and FAO− group versus placebo, the FAO+ group demonstrated a greater benefit in terms of improving FEV_1_ (0.159 L versus 0.103 L) and a more significant reduction of exacerbations leading to emergency department visits and/or hospitalization, compared to the FAO– group [75]. Collectively, based on the fact that FAO is more prevalent in severe asthmatics, treatment with monoclonals according to their indications might be extremely beneficial for these patients compared to standard care with bronchodilators and/or corticosteroids.

The oldest mAb available for severe asthmatics is the anti-IgE biologic omalizumab. It has been used for almost two decades now, with excellent results in severe allergic asthma. Its mandatory prerequisite is the presence of a positive skin prick test to a perennial allergen [76]. Its efficacy in FAO was recently estimated in a post hoc analysis of EXTRA trial, where severe asthmatics were randomized to omalizumab or placebo for a duration of 48 weeks. Patients were split into subgroups based on two key factors. Bronchodilator responsiveness (BDR), which was defined as a >12% change in FEV_1_ post bronchodilation and FAO presence based on an FEV_1_/FVC ratio >70% or not. Omalizumab reduced the exacerbation rate on patients with high BDR whether they were FAO+ or FAO–. The results, however, were not as potent in the low BDR group, with the trend of AER slightly in favor of omalizumab, without statistical significance though. Additionally, the only subgroup with evidence of improvement in lung function was the high BDR FAO– one, which is the group having both favorable factors [77]. These results showcase the existence of a difficult to treat phenotype with low BDR, especially if it co-exists with FAO, where omalizumab does not seem to be effective.

The newest mAb approved for severe asthma is the double inhibitor of both IL-4 and IL-13 dupilumab. Although its use has just been implemented in clinical practice, early indications from the Liberty Asthma QUEST RCT show that patients with fixed obstruction could possibly benefit. In this trial dupilumab 200 mg or 300 mg every 2 weeks compared to placebo exhibited significant amelioration of lung function parameters across both large and small airways in measurements such as FEV_1_, FEV_1_/FVC and FEF_25–75%_ [78]. These data are promising and further investigation of severe asthmatics with non-reversible airflow obstruction is required in the near future.

A novel potential target of biological treatment in T2 high inflammation, especially in the setting of FAO, could be interleukin-25 (IL-25). Recent research has shown that fibrocytes expressing IL-25 receptor (IL-25R+) inside peripheral mononuclear cells are increased in asthmatics compared to healthy controls. Additionally, a number of these IL-25R+ fibrocytes was positively correlated with the severity of constant airflow obstruction, more specifically with the reduction of the FEV_1_/FVC ratio and the reduction of FEV_1_ [79]. IL-25 is expressed by bronchial epithelial cells and has been implicated in airway inflammation and remodeling [80]. Whether the axis of IL-25, fibrocytes and FAO is of clinical significance remains a dilemma that trials could, should and hopefully will answer in the near future.

## 8. The Distinct “ACO” Phenotype-Asthma and COPD Co-Exist

This distinct phenotype, which comprises characteristics of both asthma and COPD, raises several therapeutic questions. Since patients with this phenotype are most commonly excluded from all the randomized trials of both asthma and COPD management, data are scarce as to the optimal treatment regimen. The GOLD 2021 update strongly recommends that patients exhibiting these characteristics should no longer be labeled as “ACO” and instead should be treated in a personalized manner, since treatment response may vary significantly. Moreover, it is noted that in such cases, “pharmacotherapy should primarily follow asthma guidelines” [81]. Current evidence suggests that a combination of medications used for asthma and COPD demonstrates an adequate response in this subgroup, especially if clinicians follow a treatable-trait oriented approach [82].

ICS clearly have a role to play in the treatment of the “ACO” phenotype. In a trial that compared the use of double therapy with ICS/LABA for a 3-month period in 152 patients with either ACO or COPD-alone, the amelioration in lung function was greater in the ACO group (*p* = 0.002), showing that ICS is potent in this mixed asthma-COPD phenotype. It should be noted, however, that in the subgroup of patients with severe to very severe airflow limitation (FEV_1_ < 50% predicted), no improvement was observed, possibly highlighting the efficacy of ICS in earlier stages of the disease [83].

A retrospective analysis using Taiwan’s National Health Insurance Research Database, with a mean follow up period of 10 years, demonstrated that use of LAMA or ICS/LABA combination in patients with ACO versus monotherapy with either ICS or LABA is beneficial in terms of exacerbation risk. The Hazard Ratio (HR) of the LAMA group was 0.51 (95% confidence interval (CI) 0.49–0.54), while the HR of the ICS/LABA group was 0.61 (95% CI 0.60–0.62) [84].

The use of LAMA is the backbone of COPD treatment for years. Recently, the addition of tiotropium in difficult-to-treat asthma, despite the proper use of an ICS/LABA combination, has proved to effectively reduce the chance of exacerbation, while improving FEV_1_ as well. This effect was well established in two replicate RCTs, including 912 asthmatics under ICS/LABA treatment that randomized patients to either tiotropium 5 mg or placebo for 48 weeks [85]. The addition of tiotropium is also a first-line choice for the neutrophilic endotype of FAO+ severe asthma, according to these results.

Furthermore, a 12-week open label randomized study, including 17 patients with ACO under fluticasone/vilanterol, assessed the addition of LAMA (umeclidinium 62.5 μg once per day). It was observed that the addition of umeclidinium in these patients managed to increase lung function by statistically significant improvements in FEV_1_ and FVC (*p* < 0.01 for both parameters) [86].

As expected, since LAMA have a more potent role in COPD, patients with asthma and co-existing emphysema may exhibit a greater benefit from the addition of tiotropium. In a double-blind randomized control trial, which compared lung function in 18 asthmatics with emphysema versus 18 without, the addition of tiotropium in the double treatment regime of ICS/LABA was observed to ameliorate lung function in both groups versus placebo. Additionally, the improvement in FEV_1_ and FVC was even greater in subjects with emphysema [87].

Conclusively, since LAMA have been officially implemented in both GOLD and GINA guidelines for COPD and asthma respectively, a direct extrapolation can also be made for the “ACO” subgroup, deeming the triple therapy with ICS/LABA/LAMA a potent therapeutic choice [1,81]. The authors suggest that personalizing treatment is the optimum choice, since results from randomized trials underline that the response is strongly based on treatable traits that can maximize the potency of a treatment regime.

However, patients with co-existing asthma and COPD under triple therapy, may still exacerbate. In such cases, there are few options for a further escalation in treatment. Such an option could be the addition of macrolides. It has been shown that in severe asthmatics who still exacerbate under high dose ICS/LABA, the addition of azithromycin 500 mg three times per week managed to exhibit a statistically significant reduction of the annual exacerbation rate compared to placebo (1.07 per patient versus 1.86 per patient, *p* < 0.001) and a concomitant improvement in quality of life, measured through the ACQ-6 questionnaire (*p* < 0.001) [88]. These findings can rationalize the use of macrolides, especially in the neutrophilic endotype and in the “ACO” subgroup, at least in a “trial and error” manner.

## 9. Conclusions

The management and medical intervention in FAO and/or ACO patients requires a holistic approach, which is not clearly established in guidelines worldwide regarding the management of major chronic diseases. This holistic approach includes FAO/ACO management based on pharmacological and non-pharmacological treatment, guideline-based management for specific co-morbidities, and modification of the risk factors: physical activity, pulmonary rehabilitation, vaccinations (flu, pneumococcal, COVID-19), smoking cessation and diet modification (Figure 1). More studies are needed to clarify and establish a holistic approach to medical intervention in FAO/ACO patients and provide clinicians with more individualized therapeutical options based on specific treatable traits.

## Figures and Tables

**Figure 1 jpm-12-00333-f001:**
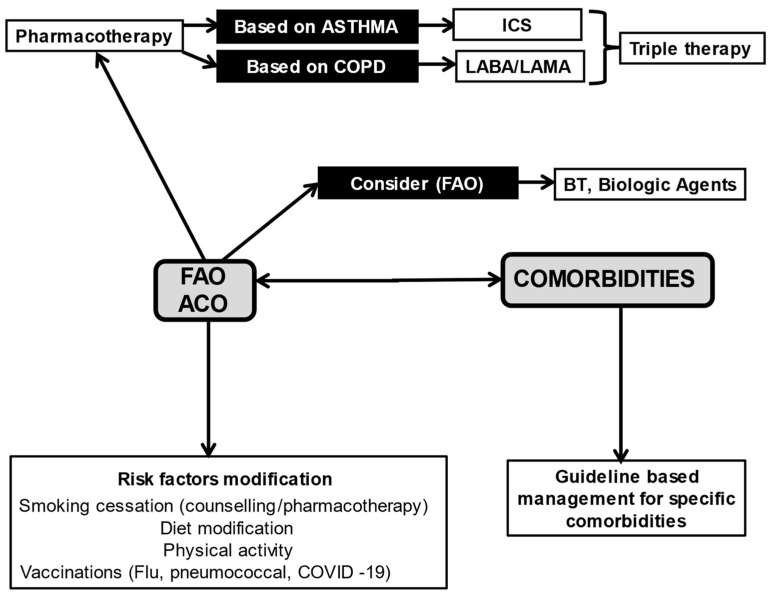
FAO asthma-ACO: A holistic approach to medical intervention. FAO = Fixed-airflow obstruction; ACO = Asthma COPD Overlap; ICS = Inhaled Corticosteroids; LABA = Long-Acting Bronchodilators; LAMA = Long-Acting Muscarinic Antagonists; BT = Bronchial Thermoplasty.

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
