# Peer review of "Asthma with Fixed Airflow Obstruction: From Fixed to Personalized Approach"

_jpm, 2022, doi:10.3390/jpm12030333_

Round 1
Reviewer 1 Report
The review offers an insight into asthma with fixed airflow obstruction and the efforts deserve recognition; however, the manuscript raises several questions and comments that need to be addressed.
- The review lacks discussion of children and adolescents with asthma with fixed airflow obstruction. Lung function impairment in childhood may influence adult outcome. The risk factors and pathophysiology of the following reference is suggested to be discussed in the manuscript.
reference: Risk factors for fixed airflow obstruction in children and adolescents with asthma: 4-Year follow-up. Pediatr Pulmonol. 2020 Mar;55(3):591-598.
- The description of potential diagnostic modalities could include more details of small airways dysfunction and the following references are suggested to be discussed in the manuscript.
references:
- Oscillometry and asthma control in patients with and without fixed airflow obstruction. J Allergy Clin Immunol Pract. 2021 Dec 31;S2213-2198(21)01453-7
- Prevalence of small airways dysfunction in asthma with- and without-fixed airflow obstruction and chronic obstructive pulmonary disease. Asian Pac J Allergy Immunol, 2021 Dec;39(4):296-303.
- Dynamic compliance and reactance in older non-smokers with asthma and fixed airflow obstruction. Eur Respir J. 2021 Aug 12;58(2):2004400
- The review does not explain the role of parenchymal destruction/emphesema in asthma with fixed airflow obstruction. This is an important issue related to pathophysiology, diagnosis and treatment. The following references are suggested to be discussed in the manuscript.
references:
- Parenchymal destruction in asthma: Fixed airflow obstruction and lung function trajectory. J Allergy Clin Immunol. 2021 Aug 24;S0091-6749(21)01302-6.
- Correlation of matrix-related airway remodeling and bradykinin B1 receptor expression with fixed airflow obstruction in severe asthma. Allergy. 2021 Jun;76(6):1886-1890.
- The review does not provide longitudinal data of clinical outcomes, such as the comparison of asthma with fixed airflow obstruction and COPD. The following reference is suggested to be discussed in the manuscript.
reference - Fixed airflow obstruction due to asthma or chronic obstructive pulmonary disease: 5-year follow-up. J Allergy Clin Immunol. 2010 Apr;125(4):830-7.
- The review does not discuss how to differentiate asthma with fixed airflow obstruction and ACO. The following reference is suggested to be discussed in the manuscript.
reference - Coexisting COPD in elderly asthma with fixed airflow limitation: Assessment by DLco %predicted and HRCT. J Asthma. 2017 Aug;54(6):606-615.
Author Response
The review offers an insight into asthma with fixed airflow obstruction and the efforts deserve recognition; however, the manuscript raises several questions and comments that need to be addressed.
- The review lacks discussion of children and adolescents with asthma with fixed airflow obstruction. Lung function impairment in childhood may influence adult outcome. The risk factors and pathophysiology of the following reference is suggested to be discussed in the manuscript.
Reference 1: Risk factors for fixed airflow obstruction in children and adolescents with asthma: 4-Year follow-up. Pediatr Pulmonol. 2020 Mar;55(3):591-598.
Answer
The reviewer is right, thank you for the comment. A new paragraph has been added in the discussion section, which now reads in red:
Although the precise risk factors for FAO in children remain mostly unknown, there is evidence they are quite similar to those observed in adults. A prospective cohort study with asthmatic children and adolescents revealed that 9.5% of them developed FAO during the 4-year follow-up. Subjects in FAO group were older than 10 years of age, had higher BMI, more severe asthma and reported higher budesonide consumption and more frequent exacerbations and hospitalizations than those in the non‐FAO group. Among the investigated characteristics, only frequent exacerbations and asthma severity categorized as steps 4 to 5 were proven independent risk factors for FAO.
- The description of potential diagnostic modalities could include more details of small airways dysfunction and the following references are suggested to be discussed in the manuscript.
Reference 2.1: Oscillometry and asthma control in patients with and without fixed airflow obstruction. J Allergy Clin Immunol Pract. 2021 Dec 31;S2213-2198(21)01453-7
Reference 2.2: Prevalence of small airways dysfunction in asthma with- and without-fixed airflow obstruction and chronic obstructive pulmonary disease. Asian Pac J Allergy Immunol, 2021 Dec;39(4):296-303.
Reference 2.3: Dynamic compliance and reactance in older non-smokers with asthma and fixed airflow obstruction. Eur Respir J. 2021 Aug 12;58(2):2004400
Answer
The reviewer is right, thank you for the comment. A new paragraph has been added in the discussion section, which now reads in red:
Currently, there is a variety of definitions of FAO used in clinical practice and medical research including use of pre- and/or post-bronchodilator spirometry values, FEV1/FVC ratio, absolute or relative cutoffs as well delineation based on the absence of a bronchodilator response alone. Spirometry predominantly assesses the flow of larger more central airways and is not a sensitive measure of the small peripheral airway dysfunction and abnormal mechanical properties of lung tissue that contribute to the pathogenesis of severe asthma. In order to assess the small airways and parenchyma in asthma, a number of different techniques have been used. Oscillometry, increasingly utilized both in clinical practice and medical research, does not require effort to force expiration that may affect small airway closure and can differentiate if an increase in the total resistance is located at central or at peripheral airways. The prevalence of increased small airway resistance using oscillometry has been found significantly higher in asthmatic patients with FAO and COPD than in asthmatics without FAO. In addition, the presence of abnormal post-bronchodilator reactance in oscillometry has been related to low asthma control in asthma patients with FAO and contributed in identifying 25% more patients with poor asthma control compared to spirometry parametres alone.
- The review does not explain the role of parenchymal destruction/emphesema in asthma with fixed airflow obstruction. This is an important issue related to pathophysiology, diagnosis and treatment. The following references are suggested to be discussed in the manuscript.
Reference 3.1: Parenchymal destruction in asthma: Fixed airflow obstruction and lung function trajectory. J Allergy Clin Immunol. 2021 Aug 24;S0091-6749(21)01302-6.
Reference 3.2: Correlation of matrix-related airway remodeling and bradykinin B1 receptor expression with fixed airflow obstruction in severe asthma. Allergy. 2021 Jun;76(6):1886-1890.
Answer
The reviewer is right, thank you for the comment. A new paragraph has been added in the discussion section, which now reads in red:
Although eosinophils and CD4+ lymphocytes seem to play important roles in directing and maintaining the inflammatory cascade, the association between remodeling and inflammation is still unclear. Allergen-induced myofibroblasts in the airways of asthmatic patients have been found to persist even after the dissolution of inflammatory infiltrates. Mediators like bradykinin may provoke acute inflammatory effects in the airways, including induction of fibroblasts to differentiate into α-smooth muscle actin (α-SMA)+ myofibroblasts and release proangiogenic factors, thus contributing to airflow obstruction in chronic asthma. Patients with severe asthma demonstrated high expression of bradykinin receptors B1 and B2 (B1R, B2R) in their bronchial submucosa, in the α-SMA+ mesenchymal cells of the bronchial mucosa and in their epithelium as well. Interestingly, α-SMA and B1R were markers associated with impaired pulmonary function and development of FAO.
- The review does not provide longitudinal data of clinical outcomes, such as the comparison of asthma with fixed airflow obstruction and COPD. The following reference is suggested to be discussed in the manuscript.
Reference 4: Fixed airflow obstruction due to asthma or chronic obstructive pulmonary disease: 5-year follow-up. J Allergy Clin Immunol. 2010 Apr;125(4):830-7.
Answer
The reviewer is right, thank you for the comment. A new paragraph has been added in the discussion section, which now reads in red:
The natural course of the disease and 5-year longitudinal data of clinical outcomes reveal that asthma patients with FAO exhibiting a progressive postbronchodilator FEV1 decline over time with a rate similar to this of patients with same degree of FAO at the beginning of follow-up due to COPD. The mean number of exacerbations per patient-year was no different between patients with FAO due to asthma or COPD and it was significantly higher compared with that seen in the asthmatic patients without FAO. In addition, sputum eosinophils at baseline were found to corelate positively with the frequency of exacerbation in patients with FAO due to asthma.
- The review does not discuss how to differentiate asthma with fixed airflow obstruction and ACO. The following reference is suggested to be discussed in the manuscript.
Reference 5: Coexisting COPD in elderly asthma with fixed airflow limitation: Assessment by DLco %predicted and HRCT. J Asthma. 2017 Aug;54(6):606-615.
Answer
The reviewer is right, thank you for the comment. A new paragraph has been added in the discussion section, which now reads in red:
ACO diagnosis should involve objective and specialized examinations, in order to distinguish between asthma and COPD. Lung function tests such as diffusing capacity of the lung for carbon monoxide (DLco), imaging such as high-resolution computed tomography (HRCT) and inflammatory biomarkers such as specific-immunoglobulin E (IgE) and fractional exhaled nitric oxide (FeNO) have been widely used for this purpose. In a study that assessed the prevalence of both FAO and COPD components in elderly asthma patients, DLco %predicted <80% and the appearance of low attenuation area as a percentage (LAA%) in HRCT were used as COPD markers. The prevalence of asthma patients with FAO increased with age and as a whole, 45% of the patients older than 50 years of age showed FAO. Nearly half of asthmatics had findings compatible with coexisting COPD, with much higher prevalence in ex- and current smokers than in never smokers, despite the almost same prevalence of FAO. These findings show that smoking plays a major role in ACO development, especially in male patients, and can aid clinicians distinguish between asthma with FAO and ACO.
Reviewer 2 Report
The paper “Asthma with Fixed Airflow Obstruction: From Fixed to Personalized Approach” by Bakakos et al. is a narrative review dealing with the problem of fixed airway obstruction in asthma and its possible importance in the asthma/COPD overlap. The review is interesting and describes a not well-known problem of asthma pathophysiology and treatment.
There are some issues I would like to address to the authors.
Major issues:
- In line 49-54 the authors describe that in asthma, airway inflammation is associated with CD4+ T lymphocytes and eosinophilia as opposed to COPD where CD8+ T lymphocytes, macrophages and neutrophils are involved. This phenomenon occurs in case of allergic/atopic asthma but what about the non-atopic/non-eosinophilic/mixed form of the disease? Could the authors comment on this issue? Only from line 160 on, non-eosinophilic asthma is briefly described, should not it be mentioned a bit earlier?
- From line 146-147 on, the authors focus on allergic (eosinophilic) asthma but write that “atopic status has been estimated less common in asthmatics with FAO although the correlation between eosinophilic markers and FAO was still present.” Could the authors comment on it or write it in a clearer way?
- Could the authors comment or provide some information (if available) about the relationship between non-Th2 asthma and FAO?
Minor issues/details:
- From line 57 on, please consider to provide the reference number directly after the name of the author (like after Fabbri et al.) instead of at the end of the sentence. This would make the text easier to read.
- Check if in line 72 the word “as” is necessary.
- From line 117 on, shouldn’t the subtypes of asthma be described as Th2 and non-Th2 instead of “T2” and “non-T2” as it is in other papers dealing with asthma?
- Unify the number “1” in or not in subscript in “FEV1” in the entire text.
- Put the dot from line 362 to 361.
- In line 374 and 381 “[70]” and “[71]” shouldn’t be in subscript.
- Check carefully spelling in the entire text having in mind that in the English language dots are used for decimal numbers and not commas (see line 418 “62.5” and not “62,5” or line 421 “p<0.01” and not “p<0,01” etc.).
Author Response
The paper “Asthma with Fixed Airflow Obstruction: From Fixed to Personalized Approach” by Bakakos et al. is a narrative review dealing with the problem of fixed airway obstruction in asthma and its possible importance in the asthma/COPD overlap. The review is interesting and describes a not well-known problem of asthma pathophysiology and treatment.
There are some issues I would like to address to the authors.
Major issues:
- In line 49-54 the authors describe that in asthma, airway inflammation is associated with CD4+ T lymphocytes and eosinophilia as opposed to COPD where CD8+ T lymphocytes, macrophages and neutrophils are involved. This phenomenon occurs in case of allergic/atopic asthma but what about the non-atopic/non-eosinophilic/mixed form of the disease? Could the authors comment on this issue? Only from line 160 on, non-eosinophilic asthma is briefly described, should not it be mentioned a bit earlier?
Answer: The reviewer is right, thank you for the comment. An additional sentence regarding the neutrophilic type of asthma has been added in red. The sentence now reads as follows:
Asthma has also been associated with neutrophilic inflammation. Neutrophilic inflammation in asthma is less common and is mostly driven by lifestyle or environmental factors, such as obesity, smoking, late age of onset and occupational exposures. - From line 146-147 on, the authors focus on allergic (eosinophilic) asthma but write that “atopic status has been estimated less common in asthmatics with FAO although the correlation between eosinophilic markers and FAO was still present.” Could the authors comment on it or write it in a clearer way?
Answer: The reviewer is right, thank you for the comment. The paragraph has been rephrased in order to clarify this issue and now reads as follows in red:
On the other hand, atopic status is less common in asthmatics with FAO, although the correlation between eosinophilic markers and FAO has been strongly established. This demonstrates that the actual endotype leading to eosinophilic inflammation affects the likelihood of developing FAO. - Could the authors comment or provide some information (if available) about the relationship between non-Th2 asthma and FAO?
Answer: The reviewer is right, thank you for addressing this issue. A sentence has been added in red which reads as follows:
Although sputum eosinophilia is one of the most important determinants of the development of FAO in severe asthma, marked airway neutrophilia has been found to correlate with FAO in asthma as well. However, the exact impact of neutrophilic lung inflammation in FAO development is unclear, since other studies have not demonstrated differences in sputum neutrophils between asthmatics with and without FAO.
Minor issues/details:
- From line 57 on, please consider to provide the reference number directly after the name of the author (like after Fabbri et al.) instead of at the end of the sentence. This would make the text easier to read.
Answer: The reviewer is right, the suggestion has been implemented in the whole text. - Check if in line 72 the word “as” is necessary.
Answer: The word “to” has been removed and the sentence now reads as follows: The structural changes of the airways, broadly referred as airway remodeling….. - From line 117 on, shouldn’t the subtypes of asthma be described as Th2 and non-Th2 instead of “T2” and “non-T2” as it is in other papers dealing with asthma?
Answer: We thank the reviewer for highlighting this point. The authors have opted to use “T2 high” and “non-T2 high” instead of Th2, since it has been demonstrated that T2 high inflammation occurs not only from T-helper 2 cells but also from Innate Lymphoid Cells 2 (ILC2). Therefore, the authors feel that “Th2” would confuse the reader as it would imply that the T-helper 2 cascade is the only endotype leading to T2 inflammation.
Reference: Kato A. Group 2 Innate Lymphoid Cells in Airway Diseases. Chest. 2019 Jul;156(1):141-149. doi: 10.1016/j.chest.2019.04.101. Epub 2019 May 10. PMID: 31082387; PMCID: PMC7118243
- Unify the number “1” in or not in subscript in “FEV1” in the entire text.
Answer: We thank the reviewer for his suggestion, the change has been implemented.
- Put the dot from line 362 to 361.
Answer: We thank the reviewer for his suggestion, the change has been implemented.
- In line 374 and 381 “[70]” and “[71]” shouldn’t be in subscript.
Answer: We thank the reviewer for his suggestion, the change has been implemented.
- Check carefully spelling in the entire text having in mind that in the English language dots are used for decimal numbers and not commas (see line 418 “62.5” and not “62,5” or line 421 “p<0.01” and not “p<0,01” etc.).
Answer: We thank the reviewer for his suggestion, the change has been implemented and the text has been thoroughly checked.
Round 2
Reviewer 1 Report
The authors modified the manuscript appropriately.